# Specific Antibodies and Arachidonic Acid Mediate the Protection Induced by the *Schistosoma mansoni* Cysteine Peptidase-Based Vaccine in Mice

**DOI:** 10.3390/vaccines8040682

**Published:** 2020-11-16

**Authors:** Hatem Tallima, Marwa Abou El Dahab, Rashika El Ridi

**Affiliations:** 1Zoology Department, Faculty of Science, Cairo University, Giza 12613, Egypt; htallima@aucegypt.edu; 2Department of Chemistry, School of Science and Engineering, American University in Cairo, New Cairo 11835, Egypt; 3Zoology Department, Faculty of Science, Ein Shams University, Cairo 11566, Egypt; m_aboueldahab_78@yahoo.com

**Keywords:** *Schistosoma mansoni*, cysteine peptidases, vaccine, humoral antibodies, uric acid, arachidonic acid

## Abstract

Several reports have documented the reproducible and considerable efficacy of the cysteine peptidase-based schistosomiasis vaccine in the protection of mice and hamsters against infection with *Schistosoma mansoni* and *Schistosoma*
*haematobium*, respectively. Here, we attempt to identify and define the protection mechanism(s) of the vaccine in the outbred CD-1 mice-*S. mansoni* model. Mice were percutaneously exposed to *S. mansoni* cercariae following immunization twice with 0 or 10 μg *S. mansoni* recombinant cathepsin B1 (SmCB1) or L3 (SmCL3). They were examined at specified intervals post infection (pi) for the level of serum antibodies, uric acid, which amplifies type 2 immune responses and is an anti-oxidant, lipids, in particular, arachidonic acid (ARA), which is an endoschistosomicide and ovocide, as well as uric acid and ARA in the lung and liver. Memory IgG1, IgG2a, and IgG2b antibodies to the cysteine peptidase immunogen were detectable at and following day 17 pi. Serum, lung, and liver uric acid levels in immunized mice were higher than in naïve and unimmunized mice, likely as a consequence of cysteine peptidase-mediated catabolic activity. Increased circulating uric acid in cysteine peptidase-immunized mice was associated with elevation in the amount of ARA in lung and liver at every test interval, and in serum starting at day 17 pi. Together, the results suggest the collaboration of humoral antibodies and ARA schistosomicidal potential in the attrition of challenge *S. mansoni* (*p* < 0.0005) at the liver stage, and ARA direct parasite egg killing (*p* < 0.005). The anti-oxidant and reactive oxygen species-scavenger properties of uric acid may be responsible for the cysteine peptidase vaccine protection ceiling. This article represents a step towards clarifying the protection mechanism of the cysteine peptidase-based schistosomiasis vaccine.

## 1. Introduction

A non-adjuvanted cysteine peptidase-based vaccine represents a means of alleviating schistosomiasis symptoms and reducing transmission, as it consistently and repeatedly elicits a highly significant (*p* < 0.005, *p* < 0.0001) reduction in challenge *Schistosoma mansoni* and *Schistosoma haematobium* burden in mice and hamsters [1,2,3,4,5,6,7,8,9]. The reduction has a ceiling in the range of 50% to 65%, which is higher than the 40% benchmark set by the World Health Organization for the progression of schistosome vaccine antigens into clinical trials in non-human primates and in humans [10]. The decrease in the number of parasite eggs in the liver and intestine is not as remarkable as for the worm load. Yet, thorough intestine oogram studies and evaluations of granulomas numbers and diameter in the liver have revealed that the majority of trapped eggs are not viable, which results in reduced disease symptoms and parasite transmission [6,7,8,9]. These findings justify the candidacy of the cysteine peptidase-based vaccine for independent trials in mice, and in efforts to define its protection mechanism(s) [9].

It has been established that cysteine peptidases, interacting with epithelial, endothelial, mucosal, and innate lymphoid cells, which are upstream of the type 2 immune cascade, induce the release of polarized type 2 cytokines, alarmins, and adenosine triphosphate (ATP), and cause substantial breakdown of the epithelial barrier [11,12,13,14,15,16,17,18,19,20,21]. Cysteine peptidase-mediated catabolic activity elicits an increase in local and systemic uric acid (2,6,8 trioxypurine-C5H4N4O3) levels, subsequent to ATP degradation to adenosine, increased release of dying and dead cells’ DNA and RNA, and the conversion of adenine and guanine to uric acid [19,20,21,22,23,24,25,26,27,28,29]. Increased uric acid levels engage the inflammasome in the target and surrounding cells and amplify the type 2 immune responses [19,20,21,22,23,24,25,26,27,28,29]. As importantly, elevated uric acid levels impact liver metabolism, inducing an increase in the synthesis of phospholipids, which is necessary for cell, tissue and organ repair, including arachidonic acid (ARA) [30,31,32]. Elevated uric acid levels also interfere with ARA metabolism to eicosanoids, allowing an increase in the level of circulating free unesterified ARA [33]. Arachidonic acid, also known as 4,8,7,11 eicosatetraenoic, is an omega 6 fatty acid (C20:4 n-6) that has been shown to be a potent schistosomicide in in vitro, and in vivo trials in rodents and children [34,35,36,37,38]. More importantly, several recent studies have documented ARA endoschistosomicidal potential [9,39,40,41,42].

In the present study, we opted for outbred mice since the vaccine is destined for humans, and to concentrate on data recorded post *S. mansoni* challenge infection (pi). The humoral immune responses of unimmunized and cysteine peptidase-vaccinated mice at specified intervals pi were examined to determine whether specific antibodies to the immunogens have any role in protection in this model, and at what time-point. The levels of serum uric acid and lipids were evaluated along with immunohistochemical assessment of the levels of uric acid and ARA in the lung and liver. In addition, ARA in vitro ovocidal activity was demonstrated. The data obtained showed that vaccine protection correlates with the release of copious amounts of specific antibodies, and increases in uric acid and ARA levels in serum and tissues at and following day 17 pi, which coincides with the post-lung, liver, and post-liver parasite stages. This article proposes that specific antibodies to cysteine peptidases and the endoschistosomicide and ovocide ARA collaborate to induce parasite worms and eggs attrition, which is a novel mechanism for the anti-schistosomiasis protective capacity of the cysteine peptidase-based schistosomiasis vaccine.

## 2. Materials and Methods

### 2.1. Ethics Statement

All animal experiments were approved by the Institutional Animal Care and Use Committee (IACUC) of the Faculty of Science, Cairo University, permit numbers CUFS F PHY 21 14, CUFS-F-Imm-5-15, and CUIS 36 16.

### 2.2. Parasites and Animals

Cercariae of an Egyptian strain of *S. mansoni* were obtained from the Schistosome Biological Materials Supply Program, Theodore Bilharz Research Institute (SBSP/TBRI), and used immediately after shedding from *Biomphalaria alexandrina* snails. Female CD-1 mice, weighing 21.5 g ± 1.8, were percutaneously infected with 150 (Experiment 1) or 200 (Experiment 2) highly viable *S. mansoni* cercariae, as described previously [1,2,5,6], to check the robustness of the vaccine protective capacity.

### 2.3. Cysteine Peptidases

Enzymatically active *S. mansoni* cathepsin B1 (SmCB1) and cathepsin L3 (SmCL3) produced in methyltrophic yeast *Pichia pastoris* GS115 (Invitrogen) and PichiaPink^TM^ (Thermo Fisher) strain, respectively, as described [2,4,6,7] were a gift from John P. Dalton (Queen’s University Belfast, Northern Ireland, UK).

### 2.4. Experimental Design

In each of two independent experiments, from a total of eighty female CD-1 mice, five were left unimmunized and uninfected and considered as naïve animals. The remaining 75 mice were randomly distributed into three equal groups of 25 mice each, which were subcutaneously immunized twice, with a three week-interval, with 0 (infection control group) or 10 μg SmCB1 or SmCL3. Infection control and vaccinated mice were percutaneously exposed to 150 (Experiment 1) or 200 (Experiment 2) cercariae of *S*. *mansoni* three weeks after the second immunization. A large number of challenge cercariae were used to ascertain the robustness of the cysteine peptidases-mediated protective capacity. Mice (3 per group/experiment) were euthanized with an intraperitoneal injection of 5 mg/kg thiopental sodium (EPICO, 10th of Ramadan City, Egypt), and examined individually at 0 (before infection), 3 (parasite skin stage), 10 (lung-stage), 17 (liver-stage), 24 (post-liver stage) and 40 (adult, egg-laying stage) days post infection (pi) [1,2,3,4,5,6,7,8,9] for the level of immunogen-specific antibodies, uric acid, cholesterol, total neutral triglycerides, free fatty acids, and ARA in serum, and uric acid and ARA in lung and liver, in parallel with naïve mice. Since similar cysteine peptidase effects on challenge worm burden were recorded in the two experiments (Table 1, Appendix A), parasitological, immunological, and biochemical parameters obtained in the two experiments were combined or represented as typical of the two experiments.

### 2.5. Serum and Organ Collection

Lung, liver and/or blood samples were obtained from individual naïve mice and unimmunized and immunized mice (3/group per experiment) following *S*. *mansoni* infection as previously described [9,40]. Sera were separated at 4 °C, and stored at −20 °C.

### 2.6. Parasitological Parameters

Worm burden was evaluated in individual mice 40 days pi following hepatic perfusion as described [1,2,5]. Eggs were collected from the liver and intestine following treatment with 4% KOH [1,2,5]. The percent change was calculated using the formula = (mean number in unimmunized infected mice − mean number in immunized, infected mice/mean number in unimmunized infected mice) × 100. The percentages of eggs at different developmental stages were evaluated using 3–5 fragments of the ileum. For each fragment, up to 100 eggs were counted and classified according to their developmental stage as immature, viable eggs; mature, viable eggs; and non-viable calcified eggs as described in [6,7,9]. Liver paraffin sections from each control and test mouse were stained with hematoxylin and eosin (Appendix A) and examined for the number and diameter of granulomas surrounding eggs. The granuloma number and diameter in the liver were evaluated in 3 fields per each of 3 sections for each of 6 mice per group [6,7,9]

### 2.7. Immunologic Assays

Release of mouse IL-1β, IL-12, (ELISA MAX™ Set, BioLegend, San Diego, CA, USA), and IL-13 (DuoSet ELISA Development System, R&D System, Abingdon, UK) was measured in duplicate serum samples by capture enzyme-linked immunosorbent assay (ELISA), following the manufacturer’s instructions. At every time-point pi, individual mouse sera, as well as the pooled equal volumes of sera of each mouse group were used in parallel to estimate the level of IgM and IgG class antibodies (sera diluted 1:200), and IgE and IgA antibodies (sera diluted 1:25) binding to SmCB1 or SmCL3 (250 ng/well), in duplicates, in 2 independent ELISA assays. Alkaline phosphatase-labeled monoclonal antibodies to immunoglobulin class specific heavy chain were obtained from Pharmingen (San Diego, CA, USA) or BioLegend. The results were read spectrophotometrically (Multiskan EX, Labsystems, Helsinki, Finland), and statistically analyzed using ANOVA test.

### 2.8. Serum Uric Acid and Lipid Assays

Serum samples were assessed on an individual mouse basis in duplicate, for enzymatic colorimetric determination of uric acid, total cholesterol, triglycerides, and free fatty acids following the manufacturer’s instructions, using Abcam (Cambridge, MA, USA; Uric Acid Assay Kit, ab65344,), CHRONOLAB SYSTEMS S.L. (Barcelona, Spain; Cholesterol-LQ; Triglycerides), and Sigma-Aldrich (St. Louis, MO, USA, Free Fatty Acid Quantitation Kit, MAK044), respectively. Levels of circulating unbound, free ARA were evaluated on an individual mouse basis, in duplicate or quadruplicate wells, by competitive ELISA using AA (Arachidonic Acid) ELISA Kit (E-EL-0051, Elabscience Biotechnology Co., Ltd., WuHan, China) following the manufacturer’s instructions. Absorbance readings of the uric acid, free fatty acids and ARA standard dilutions were plotted versus concentration values using Excel scatter graph and formula and serum sample concentrations and were expressed as mg/dL, nmole/mL and ng/mL, respectively.

### 2.9. Immunohistochemistry

Cryostat frozen lung and liver sections were incubated with RPMI medium supplemented with 5% fetal calf serum (RPMI/FCS) for blocking non-specific sites for 30 min and overnight at 10 °C with 100 μL RPMI/FCS containing 0 (negative controls) or 2 μg rabbit polyclonal antibody to uric acid (ab53000, Abcam) or ARA (MBS2003715, MyBioSource, San Diego, CA, USA) for 1 h at room temperature. Sections were washed with Dulbecco’s phosphate-buffered saline and incubated with 0.5 μg/100 μL goat anti-rabbit immunoglobulins labeled with alkaline phosphatase [Goat F(ab’)^2^ Anti-Rabbit IgG-H&L (AP), pre-adsorbed, Abcam] in RPMI/FCS for 1 h at room temperature. After thorough washing in 10 mM Tris-HCl/150 mM NaCl, the reaction was visualized with the Histomark RED Phosphatase Substrate Kit from Kirkegaard and Perry Laboratories (Gaithersburg, MD, USA). Photographs were acquired by light microscopy (Olympus, Tokyo, Japan).

### 2.10. Arachidonic Acid In Vitro Ovocidal Potential

Two independent experiments were performed to evaluate the effect of free, unesterified ARA on egg hatchability and miracidium viability. Eggs were mechanically collected, isolated and purified from the liver and intestine of cohorts of female CD-1 mice, infected 7 weeks previously with 200 cercariae of *S*. *mansoni* [43]. Samples of 1000 eggs were incubated in the dark, and in a humidified atmosphere for 3 h at 37 °C without CO_2_ in 1.0 mL fetal calf serum-free RPMI medium supplemented with 1% dimethyl sulfoxide (DMSO, Sigma-Aldrich) and 500, 250, 125, 60, 30 and 15 μM pure ARA (Cayman Chemical, Neratovice, Czech Republic). Following sedimentation, the eggs were washed in 1.0 mL RPMI medium and inspected by 40× light microscopy. Thereafter, eggs were suspended in 1.0 mL deionized sterile water and exposed to direct light for 1 h. The number and percent of eggs that failed to hatch and live and dead miracidia were evaluated before and after exposure to water and light.

### 2.11. Statistical Analysis

All values were tested for normality. ANOVA, Student’s t-two-tailed and/or Mann–Whitney tests were used to analyze the statistical significance of differences between selected values and were considered significant at *p* < 0.05 using GraphPad Prism version 6.04 for Windows (GraphPad Software, La Jolla, CA, USA)

## 3. Results

### 3.1. Parasitological Data

In two independent experiments, SmCB1 and SmCL3 elicited a highly significant (*p* < 0.005) reduction in total worm burden (Table 1, Appendix A). The highly significant decrease in worm burden of vaccinated hosts was not associated with a reduction in parasite egg counts in the liver and small intestine at day 40 pi (Table 1), perhaps because of the type 2 axis-mediated, transient increase in parasite fecundity [1,2,3]. The majority of eggs counted were perhaps not viable as immunization with the cysteine peptidases elicited death of the majority of ova (>70%, *p* < 0.005–0.0005, Student’s t test and *p* < 0.01, Mann–Whitney test) in the small intestine. Moreover, we found a highly significant (Mann–Whitney test) decrease in the number (*p* = 0.0079) and diameter (*p* < 0.0001) of circumoval granulomas in liver (Table 1).

The parameter changes are typical of two independent experiments. Vaccinated mice were challenged 3 weeks after the second immunization with 200 cercariae of *S. mansoni* in parallel with unimmunized mice (infected controls), and assessed (six per group) for worm burden and total liver and small intestine egg counts 40 days post infection. ANOVA, Student’s t two-tailed and Mann–Whitney tests were used to analyze the statistical significance of differences between selected values, and considered significant at *p* < 0.05. The significance of the differences between immunized and unimmunized control mice are shown above. There was no significant difference between SmCB1 and SmCL3 immunogens’ impact on any parasitological parameter tested, except for the percent of immature ova (*p* < 0.05).

### 3.2. Serum Immunological Data

Assays of serum samples by capture ELISA revealed no detectable levels of IL-1 or IL-13 in sera of naïve or infection controls, or in SmCB1- and SmCL3-immunized mice at any time-point following *S. mansoni* infection. Detectable levels of IL-12 were recorded only in sera of SmCB1-immunized mice at 7 and 24 days after challenge *S. mansoni* infection (mean of 3 mice per group ± SD = 132 ± 28 and 580 ± 20 pg/mL, respectively).

Figure 1A–H illustrates the typical humoral immune responses of infection control, and SmCB1-and SmCL3-immunized mice at 3, 10, 17, 24, and 40 days pi. The data recorded at day 3 and 10 pi corroborate our previous results, which documented limited antibody response to SmCB1 and SmCL3 before and up to two weeks after challenge infection [6], with no significant differences between the three mouse groups.

At and following day 17 pi, at the liver and post-liver parasite stage, SmCB1 released from the juvenile worms elicited memory responses in SmCB1-immunized mice whereby serum IgM, IgG1, IgG2a, and IgG2b antibody binding to the immunogen were significantly (*p* < 0.05–*p* < 0.005) higher than infection control mice. Additionally, IgA antibodies to SmCB1 were detected only in serum pooled from SmCB1-immunized mice on day 17, 24, and 40 pi. Copious amounts of anti-SmCB1 recorded reflect powerful SmCB1 immunogenicity and antibody binding, and cross-reactivity with SmCL3 (Figure 1A,C,E,F). Antibodies to SmCB1 of the IgE isotype were not detected in any serum pool diluted 1:25.

SmCL3 released from juvenile worms elicited evident memory responses as levels of serum IgM, IgG1, IgG2a and IgG2b antibodies binding to SmCL3 were significantly (*p* < 0.05) higher than infection control and SmCB1-immunized mice on day 24 and 40 pi. These results reflect SmCL3 limited immunogenicity and in vitro antibody binding. However, the poor immunogenicity of worm-derived, and not only recombinant, SmCL3 [1,3,43] was reflected in the negligible humoral responses of infection control and SmCB1-immunized mice at all time-points pi (Figure 1B,D,F,H). Antibodies to SmCL3 of the IgA or IgE isotype were not detected in any serum pool diluted 1:25.

### 3.3. Serum Uric Acid Levels

Serum uric acid levels in unimmunized infected mice showed no differences compared to the naïve controls (Day 0), except for a highly significant (*p* = 0.005) increase at 40 days pi (Figure 2A). Compared to naïve and infected control mice, immunization with SmCB1 and SmCL3 elicited an increase in serum uric acid at every interval pi (reaching a significance of *p* < 0.05, on day 17 and 24, at the parasite-liver and post-liver stage), except on day 40 (Figure 2A). The increase in uric acid levels in unimmunized mice on day 40 is likely because there was a higher number of worms compared to immunized mice.

Levels of free, unesterified ARA also fell below the values recorded in entirely naïve mice (Day 0) in all mouse groups at 3, and 10 days pi, but increased thereafter. Serum ARA levels were significantly (*p* < 0.05) higher than infection controls in the cysteine peptidase-immunized mouse groups on day 17 pi (Figure 2B). Of note is the remarkable decrease in serum levels of cholesterol, neutral triglycerides, and free fatty acids versus the increase in ARA levels in the three mouse groups on day 40 pi.

### 3.4. Serum Lipid Analyses

Serum cholesterol, total neutral triglycerides, and free fatty acid levels decreased in unimmunized mice immediately after infection and remained lower than in naïve mice (Day 0) until the end of the experiment, with the sharpest decrease being recorded for free fatty acids on day 40 (Appendix A). Immunization with SmCB1 or SmCL3 failed to significantly alter the levels of serum cholesterol, triglycerides, and free fatty acids in comparison to unimmunized mice exposed to *S. mansoni* (Appendix A).

### 3.5. Lung and Liver Immunohistochemistry

In each and every assay, lung and liver sections of naïve mice and immunized mice were incubated in the presence of irrelevant antibody and uniformly found to be negative. Examination of lung sections of SmCB1- or SmCL3-immunized mice at 10 days pi revealed an increase in uric acid content when compared to their naïve and infected control counterparts (Figure 3, upper panel). A substantial elevation in uric acid level was recorded in the liver of cysteine peptidase-immunized mice as compared to infected controls at day 17, and 24 pi (Figure 3, lower panel).

The levels of ARA in lung and liver of SmCB1-and SmCL3-immunized mice were considerably higher compared to naïve and unimmunized infected control mice in the lung at day 10 pi (Figure 4, upper panel) and liver at 17 and 24 (Figure 4, lower panel) days pi.

### 3.6. In Vitro Schistosomicidal and Ovocidal Assays

The in vitro schistosomicidal activity of specific antibodies and ARA was previously demonstrated in [35]. Herein, two independent experiments showed that exposure of viable eggs obtained from the liver and small intestine of *S. mansoni* infected mice to 60–500 μM ARA concentrations induced a significant (*p* < 0.005) percentage of eggs to hatch before exposure to water and strong light and to release miracidia, all of which were found to be dead. Of note, exposure to ARA concentrations as low as 15 μM elicited a significant (*p* < 0.05) increase in the percentage of eggs that failed to hatch after exposure to water and light (Figure 5).

## 4. Discussion

The mechanism of the highly significant (up to *p* < 0.0001, Mann–Whitney), reproducible and consistent cysteine peptidase-mediated protection against challenge schistosome infection remains obscure. This is in view of the limited cytokine and specific antibody responses at 7 to 14 days (schistosomula lung-stage) after infection when blood flukes are vulnerable as they negotiate the convoluted and extremely thin-walled pulmonary capillaries [1,2,3,4,5,6,7,8,9]. Additionally, SmCL3 appeared to be poorly immunogenic, eliciting barely detectable type 2 polarized immune responses, and cytokine and antibody production in immunized *S. mansoni* or *S. haematobium*-challenged hosts [6,7] despite cross-reactivity with SmCB1 (31% identity at the amino acid level, ABV71063.1 and CAD44625.1, respectively). However, we found that there was an increase in serum anti-SmCB1 and anti-SmCL3 antibodies, uric acid and ARA levels, and in uric acid and ARA content in the liver of immunized mice, which was concomitant with schistosome residence in the liver sinusoids, around week three to four after challenge infection at day 17 and 24 pi. We propose that together, immune cells-activating antibodies to the cysteine peptidase immunogens and ARA mediate the elimination of juvenile worms at the liver stage. This proposal is based on: (1) Repeat examinations revealed no differences in the number of schistosomula recovered from the lung of cysteine peptidase-immunized and unimmunized infection control hosts [2,5], suggesting that parasite attrition occurs at the post-lung stage in this vaccine model. (2) It has been reported that site-dependent host responses determine when and where vaccine immunity is expressed in different vaccine models [44] and besides the lung, the liver was specifically identified as a major site of *S. mansoni* elimination [44,45]. (3) The increase in uric acid was limited but significant (*p* < 0.05) subsequent to cysteine peptidase catabolic activity and was sufficient to direct the immune responses towards the type 2 axis, increase the antibody response, and elicit an increase in free ARA release, as has been well demonstrated [19,22,23,24,25,28,29,33]. (4) The importance of antibodies in mediating the cysteine peptidase vaccine protection was demonstrated in nude mice-*S. mansoni* [9], and recently, hamster-*Ancylostoma ceylanicum* [46] models. Specific IgG1, IgG2a, and IgG2b antibodies may interact with secreted-excreted cathepsin peptidases [47,48], which are stagnant in the liver sinusoids, leading to eosinophils, basophils, mast cells, neutrophils, and macrophages arming, activation, and the release of cytotoxic and inflammatory mediators. Intense immunologic reactions would force the liver-stage schistosomes to escape via extravasation to the hepatic tissue and certain death [49]. (5) ARA is a documented schistosomicide [34,35,36,37,38] and a putative endoschistosomicide [9,39,40,41,42]. The significant (*p* < 0.05), yet limited, increase in serum ARA is not expected to directly kill juvenile worms. However, it has been reported that free ARA is readily incorporated by *S. mansoni* [50] and likely activates the parasite voltage-gated channels and enzymes, namely, tegument-associated neutral sphingomyelinase (nSMase), with subsequent exposure of the otherwise hidden surface membrane antigens to antibody-mediated effector functions, resulting in parasite death [32,34,38,51]. (6) The schistosomicidal action of antibodies, peripheral blood mononuclear cells and ARA has been documented in vitro [35]. (7) An increase in serum uric acid levels in control infected mice on day 40 pi, at the time of patency (Figure 2A), resulting from the catabolic activity of parasite-released cysteine peptidases and egg-mediated cell injury combined, was associated with an increase in serum antibody levels and ARA content in serum and the liver. However, these host parasiticidal arms are barely effective against adult worms, which are already protected in relatively thick-walled blood capillaries with the blood flow sweeping away every excretory-secretory product (ESP). The role of the cysteine peptidase-based vaccine seems to be to hasten the production of these host schistosomicidal effectors, which allows engagement with parasites still vulnerable to extravasation in the liver sinusoids and to their direct attrition, thus preceding full development of the parasite’s arsenal of protective defenses [9].

Cysteine peptidase-vaccinated mice showed an increase in serum and liver ARA levels around day 17 pi, as compared to unimmunized control mice. The accumulation of free ARA in the liver may be responsible for egg attrition immediately upon trapping, as evidenced in vitro (Figure 5). This explains the decrease in the number and size of liver and intestine granulomas (Table 1), despite the increase in challenge worm fecundity that is consistently recorded in cysteine peptidase-vaccinated mice. An increase in worm fecundity often accompanies low worm burden and is a transient sequel of type 2 immune responses to schistosome infection [1,2,3,4,5,6,7,8,9], likely due to induction of a still undefined host factor(s), which increases the fecundity of the challenge schistosomes [52,53].

Of note, compared to infection with 80 [40] or 100 [9] *S. mansoni* cercariae, exposure of CD-1 mice to 150 or 200 cercariae led to a steep decrease in cholesterol, total neutral triglycerides, and free fatty acids serum levels during the first week pi, and levels were lower than for naïve mice at every time-point including the last on day 40. The almost similar post-infection serum cholesterol, neutral triglyceride, and free fatty acid levels in unimmunized and cysteine peptidase-protected mice suggest that these lipids do not affect parasite survival or infection outcome. However. these findings demonstrate the ability of schistosomes to act as lipid-lowering agents [54], and further reveal that lipid metabolism modulation correlates with the parasite load. Similar results were recently recorded in 1898 participants in 26 Ugandan fishing communities whereby moderate and heavy *S. mansoni* infection was associated with lower triglycerides, low-density lipoprotein cholesterol, and diastolic blood pressure, and was advocated for prevention of cardiometabolic disease [55].

In cysteine peptidase-vaccinated mice, an increase in uric acid levels and type 1 and type 17-related antibodies leads to activation of eosinophils, neutrophils and macrophages, stimulation of NADPH (nicotinamide adenine dinucleotide phosphate) oxidase, and release of reactive oxygen species (ROS), which is lethal to developing and juvenile schistosomes [16]. The propensity of ARA to bind to serum albumin explains the difficulty of increasing serum ARA concentrations to a level that is schistosomicidal to juvenile or adult worms [32,34,51,56]. Arachidonic acid is a powerful NADPH oxidase activator, and may thus indirectly lead to schistosome attrition [32,51,57]. However, the powerful ability of the parasite to counteract ROS action [56], and its even stronger uric acid anti-oxidant and ROS-scavenger capacity [29] tip the balance towards parasite protection, which explains the protection ceiling of the cysteine peptidase-based schistosomiasis vaccine [1,2,3,4,5,6,7,8,9]. Accordingly, further experiments are planned to assess the impact of ROS and host and worm anti-oxidant defenses. The present article contributes to understanding the mechanisms underlying the protective potential of the cysteine peptidase schistosomiasis vaccine, and to advocating the use of ARA for safe and efficacious therapy of schistosomiasis [58].

## 5. Conclusions

Cysteine peptidase-based schistosomiasis vaccine protection was shown to correlate with the release of copious amounts of specific antibodies, and an increase in uric acid and ARA levels in serum and tissues at and following day 17, which coincides with the post-lung, liver, and post-liver stages of *S. mansoni*. Immune cells-activating antibodies to the cysteine peptidase immunogens and ARA, which is a documented schistosomicide, mediate the elimination of juvenile, still vulnerable worms, and eggs deposited after patency by the surviving worms. The study suggests that the much-coveted goal of a sterilizing schistosomiasis vaccine could be achieved via devising a cysteine peptidase-based formulation that induces type 2-dominated immune responses and elevated ARA release, but by-passes increases in the anti-oxidant uric acid levels in serum and tissues. This study highlights the need to direct attention to ARA as the schistosomicide of the future.

## Figures and Tables

**Figure 1 vaccines-08-00682-f001:**
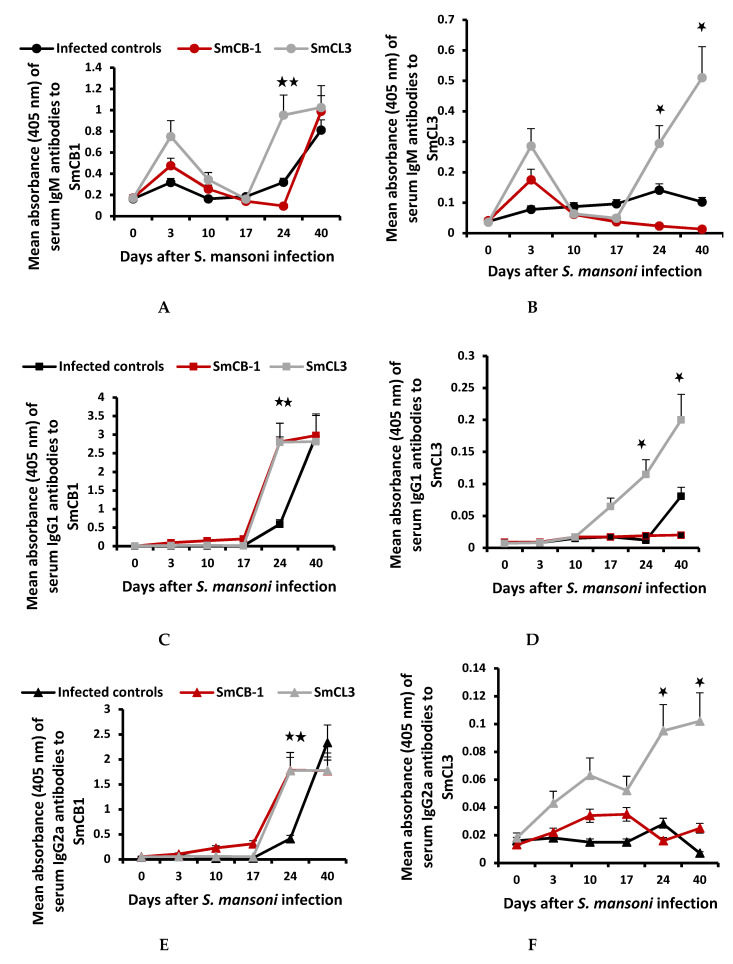
At every time-point post infection (pi), individual mouse sera, as well as, pooled equal volumes of sera of each mouse group were used, in parallel, with similar results. Mean values for the individual sera did not vary from pooled samples except for higher SD and are represented here (**A**–**H**). Antibody reactivity of individual naïve control mice to SmCB1 and SmCL3 at each time-point was negligible except for the IgM isotype, with a mean absorbance of 0.15 ± 0.05 and 0.03 ± 0.01 for SmCB1 and SmCL3, respectively. Each point represents the mean absorbance of quadruplicate wells (duplicate wells of 2 independent assays) of 1:200 diluted sera of 6 individual mice +/-SD. IgA antibodies binding to SmCB1 were only detected in 1: 25 diluted sera of SmCB1-immunized mice on day 17 (naïve, 0.081 ± 0.005; SmCB1, 0.161 ± 0.008), and days 24 and 40 (0.096 ± 0.001) pi. IgE antibodies binding to SmCB1 or SmCL3 were not detected at any time-point in any individual or pooled serum samples, diluted 1:25. ANOVA tests were used to analyze the statistical significance of differences between the three groups of mice, *p* < 0.05 (*) and < 0.005 (**).

**Figure 2 vaccines-08-00682-f002:**
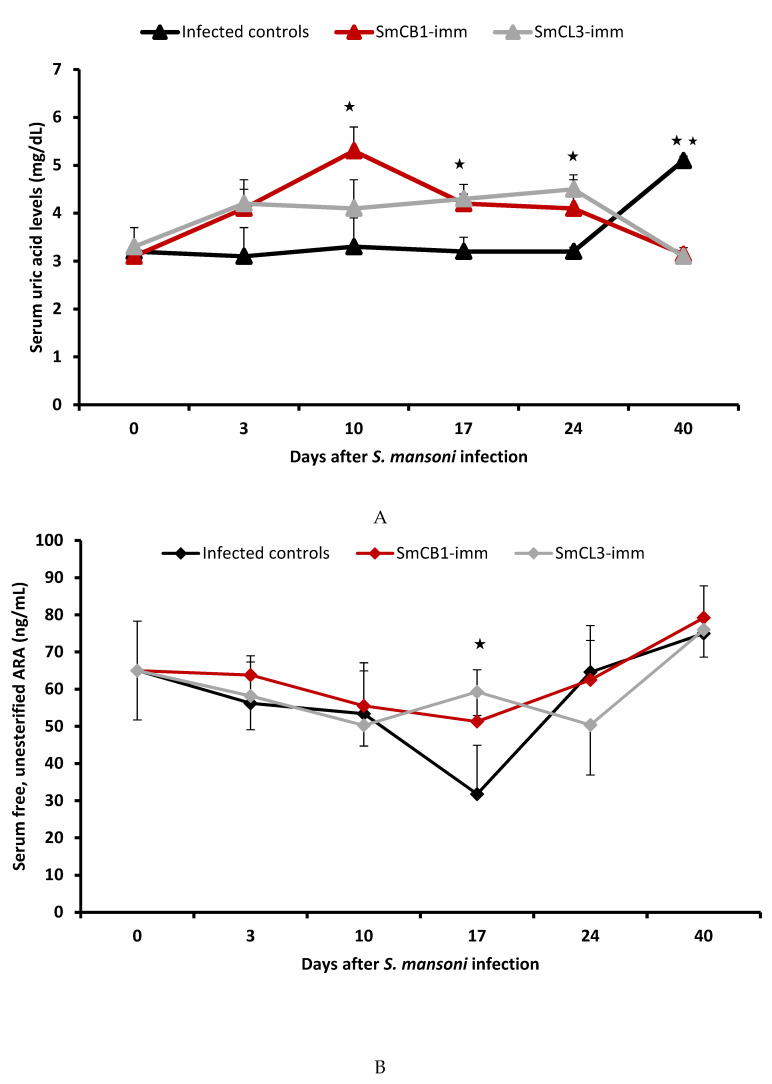
Each point represents the mean of serum uric acid (**A**) and arachidonic acid (ARA) (**B**) values for 6 individual mice +/-SD. The mean for naïve control mice = 3.2 ± 0.2 mg/mL uric acid and 60 ± 5 ng/mL ARA. All values were analyzed by ANOVA, “*t*” and Mann–Whitney tests. There was no significant difference between the impact of SmCB1 and SmCL3 immunogens. Significant differences between immunized and infected controls: *p* < 0.05 (*) and < 0.005 (**).

**Figure 3 vaccines-08-00682-f003:**
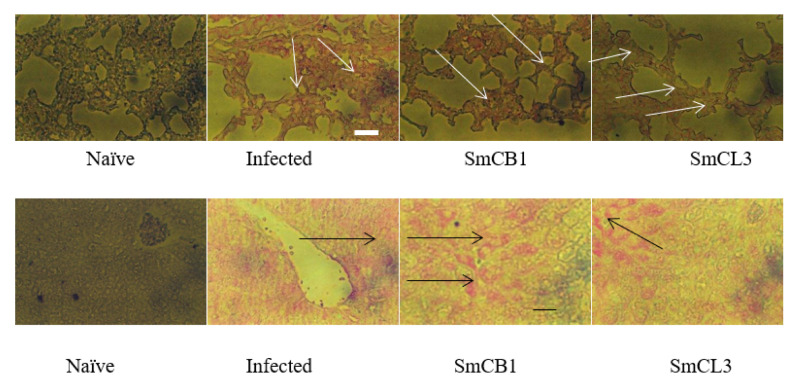
Lung (upper panel) and liver (lower panel) of each of two naïve, unimmunized (Infected) and SmCB1-(SmCB1) and SmCL3-(SmCL3) immunized mice were assayed on day 10 (upper panel) and day 17 (lower panel) post infection with irrelevant control (entirely negative, Appendix A) or anti-uric acid antibody (Ab53000, Abcam, Cambridge, MA, USA), then alkaline phosphatase-labeled antibody to rabbit immunoglobulins [Goat F(ab’)2 Anti-Rabbit IgG-H&L (AP), pre-adsorbed, Abcam] and the reaction was visualized with Histomark RED Phosphatase Substrate Kit (Kirkegaard and Perry Laboratories}. Arrows point to the areas of intense reactivity. The images shown are representative of the consistently recorded reactivity for each mouse group on day 10 (upper panel), 17, and 24 (lower panel) post *S. mansoni* infection × 200; scale bar = 20 μM.

**Figure 4 vaccines-08-00682-f004:**
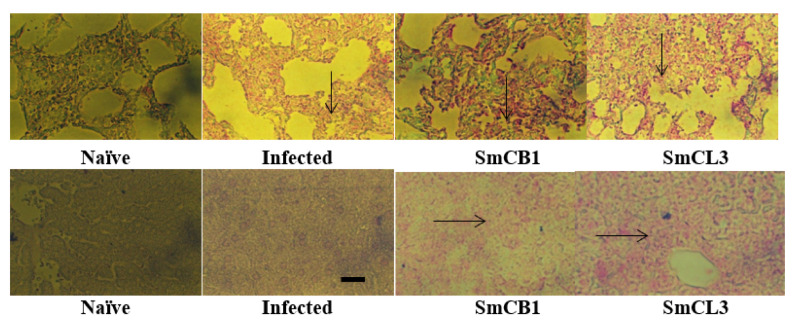
Lung (upper panel) and liver (lower panel) of each of two naïve, unimmunized (Infected) and SmCB1-(SmCB1) and SmCL3-(SmCL3) immunized mice were reacted with irrelevant control (Supplementary Material, Appendix A) or anti-ARA antibody (Ab, MBS2003715, MyBioSource, San Diego, CA, USA), then alkaline phosphatase-labeled antibody to rabbit immunoglobulins and the reaction was visualized with Histomark RED Phosphatase Substrate Kit of Kirkegaard and Perry Laboratories. The arrows point to the areas of intense reactivity. The images shown are representative of the consistently recorded reactivity for each mouse group on day 10 pi (upper panel), and day 17 and, 24 (lower panel post *S. mansoni* infection. ×200; scale bar = 20 μM.

**Figure 5 vaccines-08-00682-f005:**
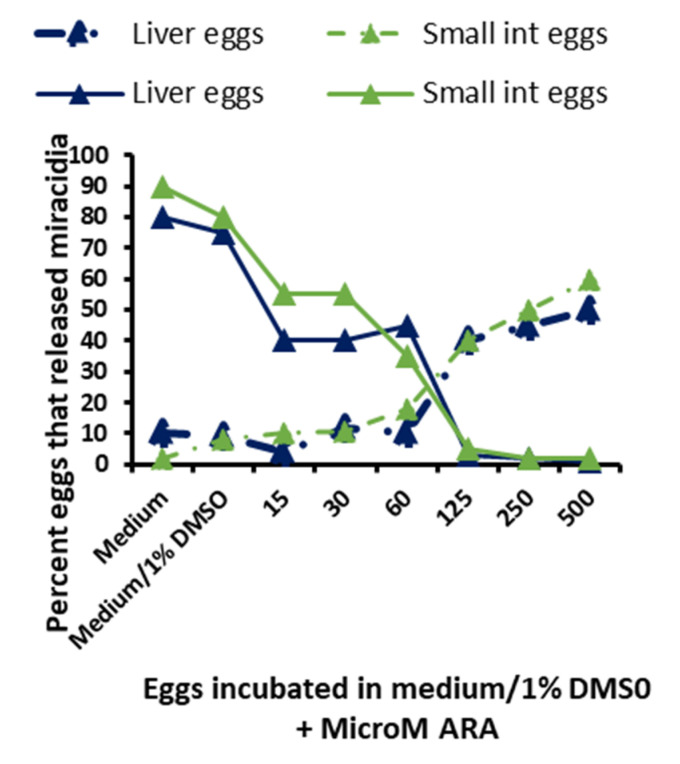
Eggs collected from the liver or small intestine (int) of mice infected with *S. mansoni* for 7 weeks were incubated for 3 h at 37 °C in RPMI medium supplemented with 1% dimehyl sulfoxide (DMSO) and 15 to 500 μM of ARA. The percentage of eggs that released miracidia before (dashed lines) or after (solid lines) 1 h exposure to water and light is the mean of 2 independent experiments. All miracidia released before water and light exposure were found to be dead while miracidia released after water and light exposure were fully viable.

**Table 1 vaccines-08-00682-t001:** Effect of immunization with SmCB1 or SmCL3 on parasitological parameters of challenge *S. mansoni* infection in outbred mice.

	VACCINE		
Parameter	Infected Controls	SmCB1-Immunized	SmCL3-Immunized
Total worm burden			
Mean ± SD	82.4 ± 3.5	36.5 ± 5.6	42.1 ± 5.8
*p* value		<0.005	<0.005
Reduction (%) *		55.7	48.9
Male worm burden			
Mean ± SD	42.8 ± 4.6	19.0 ± 3.8	22.1 ± 3.4
*p* value		<0.005	<0.005
Reduction (%)		55.6	48.6
Female worm burden			
Mean ± SD	39.2 ± 3.1	17.5 ± 2.0	20.1 ± 4.8
*p* value		<0.005	<0.005
Reduction (%)		55.3	48.7
Liver egg counts			
ean ± SD	35,800 ± 13198	37,500 ± 6442	40,666 ± 12193
*p* value Reduction (%)		NS	NS
Intestine egg counts			
Mean ± SD	29,200 ± 6610	30,500 ± 7259	29,500 ± 9354
*p* value Reduction (%)		NS	NS
% Immature ova **			
ean ± SD	41.4 ± 8.8	33.2 ± 7.7	26.2 ± 5.6
*p* value Reduction (%)		NS	0.0087 36.7
% Mature ova			
Mean ± SD	48.2 ± 13.5	26.1 ± 9.0	41.6 ± 9.0
*p* value Reduction (%)		0.012 45.8	NS
% Dead ova			
Mean ± SD	10.3 ± 5.4	40.6 ± 8.1	32.1 ± 6.5
*p* value Increase (%)		0.0043 74.6	0.0087 67.9
Granuloma number ***			
Mean ± SD	11.3 ± 0.8	7.5 ± 0.9	7.8 ± 1.1
*p* value Reduction (%)		0.0079 33.6	0.0079 31.0
Granuloma diameter			
Mean ± SD	393.0 ± 39.0	299.5 ± 55.0	279.5 ± 52.2
*p* value Reduction (%)		<0.0001 23.7	<0.0001 28.8

***** Reduction % = mean number in unimmunized mice–mean number in cysteine peptidase-immunized mice/ mean number in unimmunized mice × 100. ****** Ova developmental stages in small intestine. NS = not significant, as assessed by the Mann–Whitney test (two-tailed *p* value). ******* Granuloma number and diameter in the liver were evaluated in 3 fields per each of the 3 sections for each of the 6 mice per group.

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
