# Peer review of "Specific Antibodies and Arachidonic Acid Mediate the Protection Induced by the Schistosoma mansoni Cysteine Peptidase-Based Vaccine in Mice"

_vaccines, 2020, doi:10.3390/vaccines8040682_

Round 1

Reviewer 1 Report

In the abstract and introduction part of this manuscript, the challenges and novelty of this work is not clearly delivered.

Line 113: "Sera were separated and stored at -20C", why the separation needs to happen at -20C?

Figure 2: the statistical significance label is not very clear, suggest doing color coding to reflect the data points with significance; Also the SmCB1-imm at day 10 has a greater variation and the difference is not having stat significance?

Figure 3-4, the scale bar is super unclear, suggest using a thicker labeling to reflect the scale bar.

In discussion and conclusion: what is the significance of this work? What is the impact? How can it be applied?

Author Response

Reply to Reviewers
Reviewer 1
Comments and Suggestions for Authors
In the abstract and introduction part of this manuscript, the challenges and novelty of this work is not clearly delivered. Reply: The important comment was taken into close consideration, and changes and additions red highlighted in the Abstract and Introduction.
Line 113: "Sera were separated and stored at -20C", why the separation needs to happen at -20C? Reply: The separation may not happen at -20C, and the sentence was amended and red highlighted.
Figure 2: the statistical significance label is not very clear, suggest doing color coding to reflect the data points with significance; Also the SmCB1-imm at day 10 has a greater variation and the difference is not having stat significance? Reply: The stars were slightly enlarged and placed as to be clearly seen. Significance of SmCB1 uric acid at day 10 has been recalculated and relevant star added. We apologize for the omission.
Figure 3-4, the scale bar is super unclear, suggest using a thicker labeling to reflect the scale bar. Reply: Done.
In discussion and conclusion: what is the significance of this work? What is the impact? How can it be applied? Reply: These comments were taken into the closest consideration, and changes, additions, and a new reference [56] red highlighted.

Reviewer 2 Report

In this manuscript, Tallima and colleagues set out to assess the efficacy of two recombinant protein-based vaccines against experimental Schitosoma mansoni infection. Recombinant proteins of S. mansoni cathepsin B1 (SmCB1) and L3 (SmCL3) were generated and used to immunize outbred mice. Animals were challenged three weeks after second immunization with and S. mansoni cercariae. Different organs (lung, liver) and blood were collected to measure cytokine levels (IL-1b, IL-12, and IL-13), antigen-specific IgM and IgG levels, egg counts, worm burden, and other biochemical parameters. Mice immunized with either SmCB1 or SmCL3 showed lower total worm burdens, lower mature ova and higher dead ova numbers, and reduced number and diameter of granulomas compared to non-immunized animals. Although these results are encouraging, several points need to be addressed before the manuscript can be considered for publication:

  1. Why did the authors change the infectious dose between Experiment 1 and Experiment 2? Did they observe similar trends? Throughout the manuscript, it is not always clear if the results shown were obtained from Experiment 1 or Experiment 2 (inoculation with 150 and 200 cercariae, respectively), or if values were averaged from the two independent experiments. The authors should make this point clear.
  2. Figure 1 raises a few concerns. First, the authors need to clearly show the error bars (the value points are perhaps too big and hide the bars). Second and more importantly, the author specify that values obtained from pooled sera are shown in Figure 1 (see lines 242-243). Since these data points are presumably technical replicate (not biological replicates) from pooled sera of one experiment (not two), it is expected that SD values would be very low, and therefore the likelihood of achieving statistical significance increases. This approach can hide large variations in measured antibody titers. Ideally, the authors should show the average of titers from independent experiments or even show individual values (for each mouse).
  3. Figures 3,4, and Supplementary Figure 1 are of poor quality. The authors need to include high-resolution pictures with better color and contrast adjustments.
  4. The authors argue that higher levels of arachidonic acid (ARA) can explain the lower worm burden in immunized animals. However, this explanation raises some concerns. First, significantly different (but modest) ARA levels were observed only at Day 17 p.i. Arguably, such a small and transient difference may not be a strong driving force that explains the differences in the infection between immunized and non-immunized animals. Second, even the lowest dose of ARA tested in vitro seem to be much higher than concentration measured in vivo. Are these doses physiologically relevant? Can this have an impact in disease progression/resolution? These points need to be addressed by the authors.
  5. Figure 5B is redundant with Figure 5A (solid lines) and could be removed.
  6. On lines 316-317, the authors state that "The in vitro schistosomicidal activity of specific antibodies and ARA was previously demonstrated [35, figure 4 therein]". Did the authors mean "Figure 5 therein"? The authors should fix this mix-up.
  7. It is somewhat surprising that IL-1-beta, IL-13, and IL-12 were not detected (only IL-12 at Day 7 and 24 p.i. in the SmCB1-immunized mice) following infections. Can the authors offer an explanation why they failed to measure these cytokines?
  8. The authors suggest that higher uric acid levels in immunized mice were due to the "[...] cysteine peptidase-mediated catabolic activity" (see lines 26-27 and elsewhere). However, significantly higher uric acid levels were observed in immunized animals only at 24 days p.i. According to the methodology, animals were immunized twice with recombinant proteins, then infected 3 weeks after the last immunization. Therefore, a total of at least 45 days between the injection of the protein and detection of higher uric acid levels have gone by. Would the recombinant proteins not been completely eliminated from circulation by then, and only parasite-derived cysteine proteases be present at that time point? Furthermore, would the anti-SmCB1 and anti-SmCL3 specific antibodies (of which titers are higher immunized mice) not neutralize these proteins and block their activity? Moreover, if lower total worm burdens were observed in immunized animals, one would expect lower amounts of parasite-derived cysteine proteases to be released. How do the authors reconcile their model with the higher uric acid levels in the non-immunized animals at Day 40 p.i.? The authors should address these points in order to support their claim.

Author Response

Reply to Reviewers
Reviewer 2:
Comments and Suggestions for Authors
In this manuscript, Tallima and colleagues set out to assess the efficacy of two recombinant protein-based vaccines against experimental Schitosoma mansoni infection. Recombinant proteins of S. mansoni cathepsin B1 (SmCB1) and L3 (SmCL3) were generated and used to immunize outbred mice. Animals were challenged three weeks after second immunization with and S. mansoni cercariae. Different organs (lung, liver) and blood were collected to measure cytokine levels (IL-1b, IL-12, and IL-13), antigen-specific IgM and IgG levels, egg counts, worm burden, and other biochemical parameters. Mice immunized with either SmCB1 or SmCL3 showed lower total worm burdens, lower mature ova and higher dead ova numbers, and reduced number and diameter of granulomas compared to non-immunized animals. Although these results are encouraging, several points need to be addressed before the manuscript can be considered for publication:
1. Why did the authors change the infectious dose between Experiment 1 and Experiment 2? Did they observe similar trends? Throughout the manuscript, it is not always clear if the results shown were obtained from Experiment 1 or Experiment 2 (inoculation with 150 and 200 cercariae, respectively), or if values were averaged from the two independent experiments. The authors should make this point clear.
Reply: In our previous experiments we used 100 to 120 cercariae for challenge infection. It was aimed to test the robustness of the protection via examining the influence of increasingly higher infection doses. In all cases the reductions in worm burden in both experiments were significant at P < 0.005 (Mann-Whitney) and around 60% as shown now in Table S1. This issue is now emphasized in the Materials and Methods and yellow highlighted. Number of mice for each point is now shown in the legend to Figures 1, 2, and S2, and yellow highlighted.
2. Figure 1 raises a few concerns. First, the authors need to clearly show the error bars (the value points are perhaps too big and hide the bars). Second and more importantly, the author specify that values obtained from pooled sera are shown in Figure 1 (see lines 242-243). Since these data points are presumably technical replicate (not biological replicates) from pooled sera of one experiment (not two), it is expected that SD values would be very low, and therefore the likelihood of achieving statistical significance increases. This approach can hide large variations in measured antibody titers. Ideally, the authors should show the average of titers from independent experiments or even show individual values (for each mouse).
Reply: Figure 1 now shows values obtained from individual sera +/- SD. The value points are now reduced in size to show the SD. The changes are emphasized in the legend to Figure 1 and yellow highlighted.
3.Figures 3,4, and Supplementary Figure 1 are of poor quality. The authors need to include high-resolution pictures with better color and contrast adjustments.
Reply: Every effort was made to include high resolution pictures, including copying in PowerPoint, saving as JEPG, increasing the resolution at https://convert.town/image-dpi, with no advantage. The Figures are unmanipulated, exactly as obtained from the microscope, under exactly the same conditions for each setting.
4.The authors argue that higher levels of arachidonic acid (ARA) can explain the lower worm burden in immunized animals. However, this explanation raises some concerns. First, significantly different (but modest) ARA levels were observed only at Day 17 p.i. Arguably, such a small and transient difference may not be a strong driving force that explains the differences in the infection between immunized and non-immunized animals.
Reply: Yes, of course. The mechanism as stated in the Discussion is exposure to otherwise concealed surface membrane antigens to antibody access. We have now preceded this sentence by adding: The significant (P < 0.05), yet limited, increase in serum ARA is not expected to directly kill juvenile worms[40], lines 417-418, and yellow highlighted.
Second, even the lowest dose of ARA tested in vitro seem to be much higher than concentration measured in vivo. Are these doses physiologically relevant? Can this have an impact in disease progression/resolution? These points need to be addressed by the authors.
Reply: That is very true, and a relevant sentence, lines 459-461, and new reference where this issue was discussed at length [56] were added, yellow highlighted.
5. Figure 5B is redundant with Figure 5A (solid lines) and could be removed. Reply: Figure 5B is now removed.
6. On lines 316-317, the authors state that "The in vitro schistosomicidal activity of specific antibodies and ARA was previously demonstrated [35, figure 4 therein]". Did the authors mean "Figure 5 therein"? The authors should fix this mix-up. Reply: The mix-up was amended as figure 4 within the reference, not the article was deleted.
7. It is somewhat surprising that IL-1-beta, IL-13, and IL-12 were not detected (only IL-12 at Day 7 and 24 p.i. in the SmCB1-immunized mice) following infections. Can the authors offer an explanation why they failed to measure these cytokines? Reply: The concentrations were below the assay kit detection levels.
8. The authors suggest that higher uric acid levels in immunized mice were due to the "[...] cysteine peptidase-mediated catabolic activity" (see lines 26-27 and elsewhere). However, significantly higher uric acid levels were observed in immunized animals only at 24 days p.i. According to the methodology, animals were immunized twice with recombinant proteins, then infected 3 weeks after the last immunization. Therefore, a total of at least 45 days between the injection of the protein and detection of higher uric acid levels have gone by. Would the recombinant proteins not been completely eliminated from circulation by then, and only parasite-derived cysteine proteases be present at that time point?
Reply: It was stated lines 270-272 that: Compared to naïve and infected control mice, immunization with SmCB1 and SmCL3 elicited rise in serum uric acid at every interval pi (reaching significance, P < 0.05, on days 17 and 24, at the parasite liver- and post-liver stage).
Additionally, such uric acid increase is sufficient to skew the immune responses towards the type 2 axis, amplify the antibody responses, and increase free arachidonic release as documented in many articles [19,22-25,28,29,33], as was reported in lines 405 to 408, now yellow highlighted.
Furthermore, would the anti-SmCB1 and anti-SmCL3 specific antibodies (of which titers are higher immunized mice) not neutralize these proteins and block their activity?
Reply: High antibodies levels are detectable only after Day 17, right in time to bind to excreted-secreted host enzymes and activate effector cells as was stated in great detail, lines 411-416.
Moreover, if lower total worm burdens were observed in immunized animals, one would expect lower amounts of parasite-derived cysteine proteases to be released. How do the authors reconcile their model with the higher uric acid levels in the non-immunized animals at Day 40 p.i.? The authors should address these points in order to support their claim.
Reply: The eminent reviewer comment is very important and now gives a plausible explanation (with thanks) the higher uric acid in unimmunized versus immunized mice, as now clarified in the added and yellow highlighted sentence at the end of 3.3. Serum Uric Acid Levels, in the Results section.

This manuscript is a resubmission of an earlier submission. The following is a list of the peer review reports and author responses from that submission.

Round 1

Reviewer 1 Report

The authors attempted to investigate the mechanism for the effects of vaccination against Schistosoma Mansoni with their cathepsins B1 and L3 in mice. They verified the effects of this vaccination in some parameters including the burden of the parasites and hepatic granulomatosis lesions along with the increase of specific antibodies of IgG and IgM. They observed the increase of uric acid and arachidonic acid in plasma and in some organs so that they concluded these are the background for the mechanism for the effects of these vaccination as the increase of arachidonic acid may link the production of interleukins.

The experiments were well conducted and the data are perhaps reliable. However, they are still at the very preliminary stage and there is little mechanistic insight in the manuscript except for very speculative discussion. More experimental data are needed to support their conclusion for the mechanism of the effects of this type of vaccination.

Author Response

Response to Reviewer 1

The authors attempted to investigate the mechanism for the effects of vaccination against Schistosoma Mansoni with their cathepsins B1 and L3 in mice. They verified the effects of this vaccination in some parameters including the burden of the parasites and hepatic granulomatosis lesions along with the increase of specific antibodies of IgG and IgM. They observed the increase of uric acid and arachidonic acid in plasma and in some organs so that they concluded these are the background for the mechanism for the effects of these vaccination as the increase of arachidonic acid may link the production of interleukins.

The experiments were well conducted and the data are perhaps reliable. However, they are still at the very preliminary stage and there is little mechanistic insight in the manuscript except for very speculative discussion. More experimental data are needed to support their conclusion for the mechanism of the effects of this type of vaccination.

Reply: The article proposes a mechanism for the cysteine peptidase-based schistosomiasis vaccine reduction in challenge worm burden, impairment of worm egg viability, and decrease in granulomas number and size.  We thank the Eminent Reviewer for his remark, which has made us aware that the reason of the apparent lack of experimental data and the speculative nature of the Discussion are due to the writing.  Accordingly, the Discussion is now entirely rewritten with increased focus, shortened, and red highlighted.

Reviewer 2 Report

In this manuscript, the authors investigated the protection mechanism of the cysteine peptidase-based schistosomiasis vaccine. The authors concluded that upon immunization and challenged with Schistosoma mansoni, more antibodies against the immunogen were produced in the immunized animal, with an increase of uric acid and ARA in the serum, lung, and liver.

While this work is informative to the field of schistosomiasis vaccine, I have several concerns before it can be published. The authors are also encouraged to pay more attention to details during the preparation of the manuscript.

  1. Why did the authors conduct two immunizations instead of (the more commonly used) three immunizations? The reviewer is aware that two immunizations scheme is also used in a previous paper, reference #7 in the current manuscript. But no explanation is given.
  2. Regarding the serum immunological data in section 3.2, it would be more convincing if the specificity of the serum anti-smCB1 and anti-smCL3 activity was confirmed using a western blot analysis (using mouse serum as the primary antibody).
  3. Compared to unimmunized controls, there is a 30% increase of uric acid (from 3mg/dL to 4mg/dL), 60% increase in a single timepoint (from 3mg/dL to 5mg/dL) in immunized mice. It is difficult to conceive that the small magnitude of uric acid increase has a large impact. Although uric acid increase is significant, but not all significant changes are impactful.
  4. For SmCB-1 (the immunogen that had good immunogenicity), the authors' results didn’t necessary show that more anti-SmCB-1 antibody was produced in the immunized animals. The results only support that the anti-SmCB-1 antibody was produced EARLIER in the immunized animals.
  5. Regarding the immunohistochemistry results (Figure 3 & Figure 4), it is unclear where the positive signals are. Please add arrows pointing to the positive signals. Please also indicate any notable tissue/cells and anatomical structures in the picture.
  6. Figure 3 is lacking notations A-D, E-G, as referenced in section 3.5 of the main text.
  7. It is unclear what the authors are trying to demonstrate using the “irrelevant control” shown in SupplFig 2.
  8. Figure 2A, some y-axis labels are highlighted for unknown reasons. The axis title is also missing one parenthesis.
  9. In Figure 5 legends, 37oC should be 37℃
  10. Inconsistent fonts on multiple occasions, for example, in lines 330-331, 346-347, 402-405.

Author Response

Response to Reviewer 2.

Comments and Suggestions for Authors

In this manuscript, the authors investigated the protection mechanism of the cysteine peptidase-based schistosomiasis vaccine. The authors concluded that upon immunization and challenged with Schistosoma mansoni, more antibodies against the immunogen were produced in the immunized animal, with an increase of uric acid and ARA in the serum, lung, and liver.

While this work is informative to the field of schistosomiasis vaccine, I have several concerns before it can be published. The authors are also encouraged to pay more attention to details during the preparation of the manuscript.

  1. Why did the authors conduct two immunizations instead of (the more commonly used) three immunizations? The reviewer is aware that two immunizations scheme is also used in a previous paper, reference #7 in the current manuscript. But no explanation is given.

Reply: Two immunizations is the norm especially that the aim of every vaccine work is to translate it hurdle-free to humans.  Hence, we consistently use adjuvant-free vaccine, not more than 10 micrograms/mouse, and use the standard immunization schedule: two immunizations with a three week interval, and challenge infection three weeks after the boost immunization

  1. Regarding the serum immunological data in section 3.2, it would be more convincing if the specificity of the serum anti-smCB1 and anti-smCL3 activity was confirmed using a western blot analysis (using mouse serum as the primary antibody).

Reply: The target antigen in the indirect ELISA is ultra-pure, intact, enzymatically active cysteine peptidase, ensuring the specificity of the reaction.

  1. Compared to unimmunized controls, there is a 30% increase of uric acid (from 3mg/dL to 4mg/dL), 60% increase in a single timepoint (from 3mg/dL to 5mg/dL) in immunized mice. It is difficult to conceive that the small magnitude of uric acid increase has a large impact. Although uric acid increase is significant, but not all significant changes are impactful.

Reply: Such uric acid increase is not harmful for the animals, but is sufficient to skew the immune responses towards the type 2 axis, amplify the antibody responses, and increase free arachidonic release as documented in many articles.  This issue is now emphasized in the Discussion referring to select references, and is green highlighted.

  1. For SmCB-1 (the immunogen that had good immunogenicity), the authors' results didn’t necessary show that more anti-SmCB-1 antibody was produced in the immunized animals. The results only support that the anti-SmCB-1 antibody was produced EARLIER in the immunized animals.

Reply:  Yes, the results showed that.  This earlier response is, however, very useful, as fully developed, mature worms, residing in the mesenteric blood capillaries are refractory to antibody-mediated effector functions as noted in the Discussion, in 3 sentences at the end of the first paragraph.

  1. Regarding the immunohistochemistry results (Figure 3 & Figure 4), it is unclear where the positive signals are. Please add arrows pointing to the positive signals. Please also indicate any notable tissue/cells and anatomical structures in the picture.

Reply: Any red deposit is a positive signal, which is now emphasized by arrows.

  1. Figure 3 is lacking notations A-D, E-G, as referenced in section 3.5 of the main text.

Reply: This omission is now amended with apology and thanks.

  1. It is unclear what the authors are trying to demonstrate using the “irrelevant control” shown in SupplFig 2.

Reply: Entire negativity.  This Figure is now deleted.

  1. Figure 2A, some y-axis labels are highlighted for unknown reasons. The axis title is also missing one parenthesis.

Reply: Amended with thanks

  1. In Figure 5 legends, 37oC should be 37℃

Reply: Done with thanks

  1. Inconsistent fonts on multiple occasions, for example, in lines 330-331, 346-347, 402-405.

Reply: We do apologize.  Fonts are now amended, especially that unwanted changes occur during uploading in the site.

Reviewer 3 Report

In this study, the authors aimed to investigate the mechanism underlying the protective effect of the cysteine peptidase-based schistosomiasis vaccine in the CD-1 mice infected with S. mansoni. Mice were percutaneously exposed to S. mansoni cercariae following twice immunization with 0 or 10 mg S. mansoni recombinant cathepsin B1 (SmCB1) or L3 (SmCL3). Different parameters, including antibodies and uric acid were determined at specified intervals post infection (pi).

The study is interesting, but several concerns and drawbacks have been encountered.

  1. There is a serious issue that should be considered. Most of the data lack the results of a control (uninfected/non-immunized) group. This is very important to evaluate the impact of infection and the beneficial effects of immunization with the vaccine. Data of this group should be added to all figures.
  2. The sample size is very small, and this affects the analysis. Did the authors check normality of the results?
  3. In the legend of figure 1, the authors mentioned “Each point represents mean absorbance of quadruplicate wells (duplicate wells of 2 independent assays) of 1:200-diluted sera pooled from 3 mice per group per experiment, +/- SD. Why the authors pooled the sera of each 3 mice?
  4. The authors should show the pathological changes associated with the infection in both the lung and liver using at least H&E staining. This is very important to interpret the protective effect of immunization.
  5. A scale bar should be added to all sections.
  6. The images should be captured using the same settings. There is a clear defect in the pictures, particularly of the lung sections.
  7. The authors assayed uric acid levels and here the reviewer is curious to know the changes in the redox system. What is the impact of this vaccination on the levels of ROS and antioxidant defenses (GSH, SOD, CAT, ….. etc.).
  8. In the supplementary material, the levels of serum lipids should be provided in mg/dl not mg/ml.
  9. It is clear that the infection with S. mansoni decreases levels of serum TG, TC and FFA. Therefore, what will be the effect of this infection in patients with obesity/diabetes/metabolic syndrome where dyslipidemia is a major contributor to insulin resistance.
  10. In Figure 3, while the authors presented a lung section of the naïve group, they neglected adding a liver section from the same group. The same applies to figure 4. Also, quality of the figures should be improved.
  11. In the statistical analysis, it is not clear how the authors compared the groups after ANOVA. Also, they didn’t mention which software they used.
  12. The manuscript should be revised for the appropriate use of the English language.

Author Response

Response to Reviewer 3.

Comments and Suggestions for Authors

In this study, the authors aimed to investigate the mechanism underlying the protective effect of the cysteine peptidase-based schistosomiasis vaccine in the CD-1 mice infected with S. mansoni. Mice were percutaneously exposed to S. mansoni cercariae following twice immunization with 0 or 10 mg S. mansoni recombinant cathepsin B1 (SmCB1) or L3 (SmCL3). Different parameters, including antibodies and uric acid were determined at specified intervals post infection (pi).

The study is interesting, but several concerns and drawbacks have been encountered.

  1. There is a serious issue that should be considered. Most of the data lack the results of a control (uninfected/non-immunized) group. This is very important to evaluate the impact of infection and the beneficial effects of immunization with the vaccine. Data of this group should be added to all figures.

Reply:  Under 2.4.  Experimental Design, it was stated that "In each of two independent experiments, of a total of eighty female CD-1 mice, five were left unimmunized and uninfected and considered as NAIVE animals".   For Figures 1, and 2, and Supplementary Figure 1, values for unimmunized uninfected naïve mice are shown in Day 0, black labels for unimmunized infected controls. Sections for naïve mice are at the start of Figures 3 and 4. 

  1. The sample size is very small, and this affects the analysis. Did the authors check normality of the results?

Reply:  The analysis is mean of two independent experiments, comprising 3 mice per each group and each of six time points selected in accord with the schistosome intravascular life stages.  Yes, normality of the results was checked.

  1. In the legend of figure 1, the authors mentioned “Each point represents mean absorbance of quadruplicate wells (duplicate wells of 2 independent assays) of 1:200-diluted sera pooled from 3 mice per group per experiment, +/- SD. Why the authors pooled the sera of each 3 mice?

Reply: Under 2.7.  Immunological Assays, it was stated that " At every time-point pi, individual mouse sera and pooled equal volumes of sera of each mouse group used in parallel toestimate the…".  Similar results were obtained, and the values and SD of mean of the pools were selected to be shown in Figure 1.

  1. The authors should show the pathological changes associated with the infection in both the lung and liver using at least H&E staining. This is very important to interpret the protective effect of immunization.

Reply: As stated under 2.6 Parasitological Parameters, "Liver paraffin sections from each control and test mouse were stained with haematoxylin and eosin and examined for the number and diameter of granulomas surrounding eggs.  Granuloma number and diameter in liver were evaluated in 3 fields per each of 3 sections for each of 6 mice per group". Indicating we have very thoroughly examined and interpreted the protective effect of immunization.

  1. A scale bar should be added to all sections.

Reply: Magnifications are x 200 and x 400.  An eminent Reviewer asked to add arrows to show positive reactivity, preventing adding scale bars.

  1. The images should be captured using the same settings. There is a clear defect in the pictures, particularly of the lung sections.

Reply:  The images were captured using exactly the same settings of each experiment.

  1. The authors assayed uric acid levels and here the reviewer is curious to know the changes in the redox system. What is the impact of this vaccination on the levels of ROS and antioxidant defenses (GSH, SOD, CAT, ….. etc.).

Reply: Yes, we totally agree with the eminent Reviewer.  It is necessary to examine in depth the mechanism of the cysteine peptidase-mediated protection.  The oxidant-anti-oxidant system is instrumental in the protection and ceiling of protection obtained, as emphasized in the last sentence of the Discussion, red highlighted.

  1. In the supplementary material, the levels of serum lipids should be provided in mg/dl not mg/ml.

Reply: In our previous publications we have always illustrated serum lipids as mg/ml and need to keep it to allow comparing contrasting the findings.

  1. It is clear that the infection with S. mansoni decreases levels of serum TG, TC and FFA. Therefore, what will be the effect of this infection in patients with obesity/diabetes/metabolic syndrome where dyslipidemia is a major contributor to insulin resistance.

Reply:  The results we obtained consistently document the ability of schistosomes to act as lipid-lowering agents in mice, entirely confirming the pioneering findings of Doenhoff et al. [54].  The innovative comment of the Eminent Reviewer has a partial answer in a very recent reference [55], which is now discussed and added at the end of Discussion paragraph 3, and red highlighted

  1. In Figure 3, while the authors presented a lung section of the naïve group, they neglected adding a liver section from the same group. The same applies to figure 4. Also, quality of the figures should be improved.

Reply: An eminent reviewer disliked the entirely blank figure of lung and liver incubated in the presence of irrelevant antibody, and it was deleted.

  1. In the statistical analysis, it is not clear how the authors compared the groups after ANOVA. Also, they didn’t mention which software they used.

Reply: All values were tested for normality.  ANOVA, Students' –t- 2-tailed and/or Mann-Whitney tests were used to analyze the statistical significance of differences between selected values and considered significant at P < 0.05  The software used is now mentioned, and red highlighted.

  1. The manuscript should be revised for the appropriate use of the English language.

Reply: The manuscript was revised word by word, sentence by sentence for appropriate use of the English language.

Round 2

Reviewer 1 Report

Discussion has been improved by making statement for ARA more moderate.

Author Response

Thank you very much.

Reviewer 3 Report

The authors didn't adequately reply to the comments. The reviewer asked to add the results of the control group throughout the experiment in order to interpret the findings. However, it is clear that the authors have the data of this group at Day 0 only. This is unacceptable.

The authors confirmed that they tested the normality of their results. However, this is technically impossible with a sample size of 3. Although there are many test and the Wilk-Saphiro test is the most powerful one for testing normality of small size samples, it is difficult to apply it with this very low sample size.

Regarding the 3rd comment "Why the authors pooled the sera of each 3 mice?", the authors didn't provide any answer and only stated that they have pooled the sera of each 3 mice.

In the comment no. 5, the reviewer asked to add H&E-stained sections which are very important to interpret the protective effect of immunization. The authors replied that they performed H&E staining and examined the section, but didn't show any of them.

Comment no. 6: the scale bar could be added along with the arrows.

Comment no. 7: it is difficult to accept that the microscope settings were unified while capturing the presented images.

The authors didn't reply adequately to the comment "What is the impact of this vaccination on the levels of ROS and antioxidant defenses (GSH, SOD, CAT, ….. etc.)."

All staining panels should include sections from the control group. The authors didn't add them.

Author Response

RESPONSE TO REVIEWER

The authors didn't adequately reply to the comments. The reviewer asked to add the results of the control group throughout the experiment in order to interpret the findings. However, it is clear that the authors have the data of this group at Day 0 only. This is unacceptable.

Reply: In the past, present, and future, no assay, no test, no experiment were ever performed that do not include, start with, compare and contrast, assess, a minimum of 3 individual naïve control serum or organ samples.  The control values are now displayed in the Explanation of Figures 1 and 2 and Supplementary Figure 2, and green highlighted.

The authors confirmed that they tested the normality of their results. However, this is technically impossible with a sample size of 3. Although there are many test and the Wilk-Saphiro test is the most powerful one for testing normality of small size samples, it is difficult to apply it with this very low sample size.

Reply:  Tank you for your comments. The sample size analyzed was six not 3.

Regarding the 3rd comment "Why the authors pooled the sera of each 3 mice?", the authors didn't provide any answer and only stated that they have pooled the sera of each 3 mice.

Reply: It was stated under 2.7.  Immunological assays: "At every time-point pi, individual mouse sera, as well as, pooled equal volumes of sera of each mouse group were used, in parallel".  Similar mean values for each point were obtained.  This issue is further clarified in the Explanation of Figure 1, and green highlighted.

In the comment no. 5, the reviewer asked to add H&E-stained sections which are very important to interpret the protective effect of immunization. The authors replied that they performed H&E staining and examined the section, but didn't show any of them.

Reply: Representative sections are shown now in Supplementary Figure 1.

Comment no. 6: the scale bar could be added along with the arrows.

Reply: Scale bars are added in each Supplementary figure, and in Figures 3 and 4 as allowed by the website system.

Comment no. 7: it is difficult to accept that the microscope settings were unified while capturing the presented images.

Reply: The microscope settings were unified while capturing the presented images for each experiment.

The authors didn't reply adequately to the comment "What is the impact of this vaccination on the levels of ROS and antioxidant defenses (GSH, SOD, CAT, ….. etc.)."

Reply: This is a critical issue to be addressed very soon.  We certainly do not presently have data, but experiments are planned to examine this particular issue, which likely holds the key to the quite frustrating failure to achieve a sterilizing vaccine, and that is now added at the end of the Discussion, and green highlighted.

All staining panels should include sections from the control group. The authors didn't add them.

Reply: Sections representing liver sections from naïve mice are now added in Figures 3 and 4.